# Multi-Step Budgeted Bayesian Optimization with Unknown Evaluation Costs

**Raul Astudillo**
Cornell University
ra598@cornell.edu

**Daniel R. Jiang**
Facebook
drjiang@fb.com

**Maximilian Balandat**
Facebook
balandat@fb.com

**Eytan Bakshy**
Facebook
ebakshy@fb.com

**Peter I. Frazier**
Cornell University
pf98@cornell.edu

## Abstract

Bayesian optimization (BO) is a sample-efficient approach to optimizing costly-to-evaluate black-box functions. Most BO methods ignore how evaluation costs may vary over the optimization domain. However, these costs can be highly heterogeneous and are often unknown in advance. This occurs in many practical settings, such as hyperparameter tuning of machine learning algorithms or physics-based simulation optimization. Moreover, those few existing methods that acknowledge cost heterogeneity do not naturally accommodate a budget constraint on the total evaluation cost. This combination of unknown costs and a budget constraint introduces a new dimension to the exploration-exploitation trade-off, where learning about the cost incurs a cost itself. Existing methods do not reason about the various trade-offs of this problem in a principled way, leading often to poor performance. We formalize this claim by proving that the expected improvement and the expected improvement per unit of cost, arguably the two most widely used acquisition functions in practice, can be arbitrarily inferior with respect to the optimal non-myopic policy. To overcome the shortcomings of existing approaches, we propose the *budgeted multi-step expected improvement*, a non-myopic acquisition function that generalizes classical expected improvement to the setting of heterogeneous and unknown evaluation costs. Finally, we show that our acquisition function outperforms existing methods in a variety of synthetic and real problems.

## 1   Introduction

Bayesian optimization (BO) (Shahriari et al., 2016; Frazier, 2018) is a family of algorithms for optimizing black-box functions that performs well when the number of evaluations is limited (Snoek et al., 2012; Calandra et al., 2016; Griffiths and Hernández-Lobato, 2020). However, most BO algorithms ignore the fact that the cost of evaluating the black-box objective function may vary substantially across the optimization domain and is often unknown. Problems with this feature arise commonly in practice. For instance, in the context of hyperparameter optimization of machine learning algorithms (Swersky et al., 2013; Wu et al., 2020), certain values of hyperparameters such as the learning rate may yield longer training times. Similarly, in materials design and robotics, simulation experiments can take longer for certain parameter configurations (Field, 1999). Figure 1 illustrates heterogeneity in evaluation costs from benchmark problems used in this paper, which can vary by an order of magnitude. Failing to account for these heterogeneous evaluation costs can lead to evaluating an expensive point when another less expensive one would provide equal benefit towards finding the optimum.

35th Conference on Neural Information Processing Systems (NeurIPS 2021).

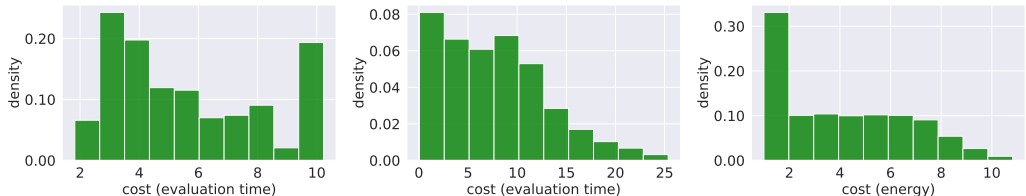

Figure 1: Evaluation times of the latent Dirichlet allocation, random forest, and energy-aware robot pushing benchmark problems (described later in Section 5.1).

We consider *budgeted* BO of a black-box objective function, whose evaluation costs are unknown and possibly heterogeneous across the domain. The goal is to find a point with the largest possible objective value by querying the objective function at a sequence of adaptively chosen points, where the total evaluation cost is subject to a budget constraint (this cost only affects evaluation, and not a point's quality upon implementation).

While some existing approaches do address heterogeneous evaluation costs, all do so heuristically, e.g. by maximizing a traditional cost-agnostic acquisition function divided by the cost of evaluation (Snoek et al., 2012; Poloczek et al., 2017; Wu et al., 2020; Lee et al., 2020b), or by rolling out a heuristic base policy (Lee et al., 2021). Importantly, most of these approaches accommodate neither budget constraints nor uncertainty about the cost function as part of the exploration-exploitation trade-off. As we argue theoretically and demonstrate through experiments, this can lead to poor performance. A notable exception is the concurrent work of Lee et al. (2021), which introduces a budget-aware non-myopic acquisition function based on rollout of a heuristic base policy. This work appeared while the present paper was under review.

**Main Contributions.** Motivated by the above shortcomings in existing work, we provide a principled approach to budgeted BO with unknown and potentially heterogeneous evaluation costs. Our main contributions are:

- We propose a Markov decision process (MDP) formulation of the budgeted BO problem with unknown and heterogeneous evaluation costs. Our formulation allows for a random time horizon (i.e., the last time before the budget is depleted), going beyond the fixed-horizon MDPs formulated in existing work on non-myopic BO.

- *Budgeted multi-step expected improvement* (B-MS-EI), a novel look-ahead acquisition function that generalizes classical expected improvement to the budgeted cost-heterogeneous setting. B-MS-EI can be seen as a principled approximation of the optimal policy of our MDP.

- We prove that expected improvement (EI) and its cost-normalized variant, two popular existing approaches, can be arbitrarily inferior with respect to the optimal non-myopic policy.

- An empirical evaluation on a number of synthetic and real-world experiments demonstrates that B-MS-EI performs favorably with respect to other acquisition functions that are widely-used in settings with heterogeneous costs.

The remainder of this work is organized as follows: In Section 2, we review related work. Our problem setup is formalized in Section 3. In Section 4, we introduce B-MS-EI and discuss its efficient maximization via one-shot multi-step Monte Carlo trees. Numerical experiments are presented in Section 5. Finally, we discuss directions of future work and conclude in Section 6.

## 2 Related Work

Our work falls within the BO framework (Frazier, 2018), whose origins date back to the seminal work of Kushner (1964). BO has been successful in a wide range of applications, such as hyperparameter tuning of machine learning algorithms (Snoek et al., 2012; Wu et al., 2020), materials design (Zhang et al., 2020), drug discovery (Griffiths and Hernández-Lobato, 2020), and robot locomotion (Calandra et al., 2016; Wang and Jegelka, 2017).

Within the BO literature, the works most closely related to ours are those that acknowledge the existence of costs for evaluating the objective function that are heterogeneous across the search space and aim to devise algorithms that are cost-aware. Much of this work has occurred in the multi-fidelity

setting (Swersky et al., 2013; Kandasamy et al., 2016, 2017; Poloczek et al., 2017; Song et al., 2019; Wu et al., 2020), i.e., where cheaper approximations of the objective function are available. The only exceptions known to us are Snoek et al. (2012), Lee et al. (2020b), and Lee et al. (2021), which consider the single-fidelity setting.

Snoek et al. (2012) proposes the *expected improvement per unit of cost (EI-PUC)*, i.e, $EI(x)/c(x)^1$ where $c(x)$ is the cost of evaluating the objective at $x$ and $EI(x)$ is the expected improvement. Lee et al. (2020b) proposes a simple variation called the *expected improvement per unit of cost with cost cooling (EI-PUC-CC)*. EI-PUC-CC is defined by $EI(x)/c(x)^\nu$, where $\nu$ is the ratio between the current remaining and initial budgets. The intuition behind the cost exponent is that evaluating points with high cost should be discouraged early in the BO loop (when $\nu \approx 1$) and accommodated as the budget is consumed and $\nu$ decreases to $0$. However, neither EI-PUC nor EI-PUC-CC consider uncertainty in the cost or measure the budget-dependent value of information in a principled way.

Work concurrent to ours, Lee et al. (2021), also tackles budgeted BO with heterogeneous costs using a non-myopic strategy. However, our work differs in both the model and solution method. Lee et al. (2021) uses a finite-horizon constrained MDP, while our model is an MDP with a random horizon. We argue that the random horizon formulation is more natural: the formulation of Lee et al. (2021) requires the addition of a zero reward, zero cost state to accommodate trajectories with a small number of evaluations. Within this formulation, Lee et al. (2021) proposes a rollout acquisition function with a particular heuristic base policy: $h - 1$ steps of EI-PUC followed by a last step of EI, where $h$ is the number of look-ahead steps performed. The acquisition function is essentially a single step of policy improvement over the "EI-PUC followed by EI" heuristic (Sutton and Barto, 2018). In contrast, our acquisition function aims to directly approximate the optimal policy.

The approach of dividing a cost-agnostic acquisition function by some cost term is widely used in practice for addressing heterogeneity in costs, and is closely related to the use of "value divided by cost" in knapsack problems (Badanidiyuru et al., 2013). In the knapsack problem, one selects items to include into a knapsack to maximize the sum of the selected items' values subject to a budget constraint on the sum of the items' costs. In this setting, myopically adding items to the knapsack that maximize value divided by cost has strong theoretical guarantees: this algorithm provides at least $1/2$ the optimal value (Williamson and Shmoys, 2011). However, this theoretical guarantee relies on the additive nature of value in the knapsack problem. In contrast, in BO, the value obtained from multiple evaluations is the maximum of the values of the evaluations, not their sum. Indeed, we show that in this setting the "value divided by cost" approach can perform arbitrarily worse than the optimal policy.

Heterogeneous evaluation costs have also been considered in the multi-armed bandits literature (Badanidiyuru et al., 2013; Xia et al., 2015, 2016). These works develop algorithms based on optimistic policies that maximize some form of reward-to-cost ratio. As mentioned above, this type of policy is sensible when the measure of performance is the cumulative regret but is not appropriate in an optimization or "best-arm identification" setting.

Our work is also closely related to non-myopic BO (Gonzalez et al., 2016; Lam et al., 2016; Yue and Kontar, 2020; Jiang et al., 2020a; Lee et al., 2020a; Jiang et al., 2020b; Lee et al., 2021), a class of acquisition functions that account for future evaluations when quantifying a point's acquisition value. To the best of our knowledge, the work of Lee et al. (2021) discussed above is the only one among these that is able to handle heterogeneous evaluation costs.

## 3 Budgeted Bayesian Optimization with Unknown Evaluation Costs

We now formally state the problem of budgeted BO with unknown evaluation costs. Given a compact optimization domain $\mathbb{X} \subset \mathbb{R}^d$, our goal is to find a point $x \in \mathbb{X}$ with the largest possible objective value by querying the black-box objective function, $f : \mathbb{X} \to \mathbb{R}$, at a sequence of points $\{x_i\}_{i=1}^n$, subject to the constraint $\sum_{i=1}^n c(x_i) \leq B$, where $c(x)$ is the cost of evaluating $f$ at $x$ and $B$ is the evaluation budget. The cost observation $c(x)$ is revealed immediately after the evaluation of $f(x)$ is performed. However, the actual cost function $c$ is unknown. As is typical in BO, we endow $f$ and $c$ with a joint prior distribution, $p$. An *observation* in our setting is a triple $(x_i, y_i, z_i) \in \mathbb{X} \times \mathbb{R} \times \mathbb{R}_{>0}$,

---

[1]This expression is for the case when $c(x)$ is known. When $c(x)$ is unknown and learned, then either the expectation is taken over the distribution of improvement to cost ratios (as we do in our experiments) or the denominator is replaced by the mean cost.

where $y_i$ is an observation of the objective $f$ at $x_i$, and $z_i$ is an observation of the cost $c$ for evaluating $f$ at $x_i$.

## 3.1   MDP Formulation

The state of our MDP at step $n$ is $\mathcal{D}_n$, defined as the set of observations so far. These sets are defined recursively by $\mathcal{D}_n = \mathcal{D}_{n-1} \cup (x_n, y_n, z_n)$ for $n \geq 1$, where $\mathcal{D}_0$ is a set of initial observations. The joint posterior distribution over $f$ and $c$ given $\mathcal{D}_n$ is denoted by $p(\cdot \mid \mathcal{D}_n)$. The *utility* generated by a particular state $\mathcal{D}_n$ is defined as the maximum observed objective value $u(\mathcal{D}_n) = \max_{(x,y,z)\in\mathcal{D}_n} y$. Note that this utility function encodes the fact that *after evaluation*, a point with maximum objective value is desired regardless of its cost. We also let $s(\mathcal{D}_n) = \sum_{(x,y,z)\in\mathcal{D}_n} z$ denote the *total cost* of observed points in $\mathcal{D}_n$.

The sets of observations $\mathcal{D}_1, \mathcal{D}_2, \ldots$ are random due to the yet unobserved values of the objective and cost functions. A *policy* $\pi = \{\pi_k\}_{k=1}^{\infty}$ is a sequence of functions, each mapping sets of observations to points in $\mathbb{X}$, so that $x_k = \pi_k(\mathcal{D}_{k-1})$. Given a set of observations $\mathcal{D}$ such that $s(\mathcal{D}) \leq B$ (i.e., there is nonnegative remaining budget), the *value function* of a policy $\pi$ is defined as $V^\pi(\mathcal{D}) = \mathbb{E}^\pi\big[u(\mathcal{D}_{N_B}) - u(\mathcal{D}_0) \mid \mathcal{D}_0 = \mathcal{D}\big]$, where the random stopping time $N_B = \sup\{k : s(\mathcal{D}_k) \leq B\}$ is the largest time step $k$ for which the budget constraint is still satisfied. The notation $\mathbb{E}^\pi[\,\cdot\,]$ indicates an expectation taken over sequences of observation sets $\mathcal{D}_1, \mathcal{D}_2, \ldots, \mathcal{D}_{N_B}$ selected by a policy $\pi$. For a set $\mathcal{D}$ where $s(\mathcal{D}) > B$ (i.e., budget is exhausted), we define $V^\pi(\mathcal{D}) = 0$. Our goal is to find a policy $\pi$ that maximizes the increase in expected utility:

$$V^*(\mathcal{D}) = \sup_{\pi \in \Pi} V^\pi(\mathcal{D}), \tag{1}$$

where $\Pi$ is the set of all possible policies. The above problem is well-defined provided that $N_B < \infty$ for all policies $\pi$. This is the case, for example, when $\ln c$ follows a Gaussian process (GP) prior, a modeling choice that we make in our numerical experiments. Since the horizon $N_B$ is random, the formulation (1) can be viewed as a stochastic shortest path problem, rather than a standard finite or discounted infinite horizon dynamic program (Bertsekas and Tsitsiklis, 1991). Note that at time $k$, the set $\mathcal{D}_k$ contains all past observed costs and fully captures the remaining budget. This formulation is capable of the following:

1. Through multi-step planning, it can navigate the trade-off of how to sequence high-cost and low-cost evaluations in order to make the best use of the given budget.

2. It can reason about uncertainty when planning optimal cost-learning. For example, an exploratory evaluation may be worthwhile in a region with moderate estimated cost and high model uncertainty: the evaluation may reveal the cost is lower than estimated, allowing us to explore the region more fully for low cost.

## 3.2   Contrast with Value-to-Cost Ratio Methods

It is not surprising that cost-agnostic methods, such as expected improvement (EI), can perform poorly when the evaluation cost is heterogeneous. In particular, ignoring cost can lead to evaluating excessively high-cost points, depleting budget and limiting future evaluations. In an attempt to avoid this, past work has focused on using a value-to-cost ratio (Snoek et al., 2012; Swersky et al., 2013; Kandasamy et al., 2016, 2017; Poloczek et al., 2017; Song et al., 2019; Wu et al., 2020; Lee et al., 2020b).

We show here, however, that value-to-cost acquisition functions exhibit a complementary kind of undesirable behavior: they may repeatedly measure excessively low-cost points that are also low-value, leading to poor overall performance. In fact, the performance can be arbitrarily bad compared with an optimal policy. Theorem 1 demonstrates this formally for the most widely-used of these policies: measuring at the point that maximizes the expected improvement per unit of cost (EI-PUC), and also for cost-agnostic expected improvement (EI).

**Theorem 1.** *The approximation ratios provided by the* EI *and* EI-PUC *policies are unbounded. That is, for any arbitrarily large $\rho > 0$ and each policy $\pi \in \{\mathrm{EI}, \mathrm{EI\text{-}PUC}\}$, there exists a Bayesian optimization problem instance (a prior probability distribution over objective and cost functions, a budget, and a set of initial observations $\mathcal{D}_0$) where $V^*(\mathcal{D}_0) > \rho\, V^\pi(\mathcal{D}_0)$.*

*Sketch of Proof.* To show that EI-PUC has an unbounded approximation ratio, the detailed proof of Theorem 1 (provided in Section A of the supplementary material) constructs a problem instance

with a prior that is independent across a discrete domain with a known cost function. There are two kinds of points: high-cost points with large prior variance; and low-cost points with low variance. To support analysis, all points have the same mean.

The variance of the low-cost points is low enough that spending the entire budget on evaluating them earns only a fraction of the value, in expectation, earned by evaluating a single high cost point. The acquisition function EI-PUC, however, does not understand this. Its greedy nature leads it to overvalue these points. Indeed, EI (the numerator of EI-PUC's acquisition function) greedily values improvement relative to the status quo, even though the status quo will likely be surpassed by other later evaluations. This overvalues small improvements like those that result from low-variance points.

EI-PUC spends its entire budget on these low-cost low-variance points, earning almost no value. In contrast, the optimal policy spends its entire budget on a single evaluation of the high-cost point, obtaining substantially more value in expectation.

For the case of EI, we construct a similar example, with the change that the low-cost points now have variance *only slightly smaller* than that of the high-cost points, while still being significantly cheaper. Here, EI performs a single evaluation of the high-cost point and runs out of budget. On the other hand, repeatedly measuring low-cost points is far superior to EI in expectation. □

Figure 2 illustrates this phenomenon in a continuous one-dimensional setting with a Gaussian process prior. Under the posterior in the first time slice, there is a lower-cost region on the left ($x < 0.5$) with low-variance and a mean that is significantly below the best point observed. There is a higher-cost region on the right ($x > 0.5$) where the mean and variance are both larger. These means and variances are such that the global maximum of the function is likely to be in the right-hand region. In its first two measurements, the EI-PUC strategy measures at two low-cost points, which results in reduction of variance in this region but (as expected) no values that contend with the likely value of the global maximum on the right. In contrast, in its first two measurements, our multi-step B-MS-EI strategy evaluates on the right, finding values that are close to the global optimum.

In this example, EI-PUC encounters the same difficulty it faces in the proof of Theorem 1: It overvalues low-cost points that improve relative to the status quo but not relative to where we hope to be near the end of the budget. While some budget remains to evaluate on the right after these evaluations, the evaluations on the left are likely unproductive toward the goal of finding a global maximum.

## 3.3 Decomposition and Truncation

We now discuss how we can additively decompose and then truncate the problem in (1) so that it is more amenable as a BO acquisition function. Let $r(\mathcal{D}_{n-1}, \mathcal{D}_n) = u(\mathcal{D}_n) - u(\mathcal{D}_{n-1})$ be the increase in utility between successive states. Using a telescoping sum and rewriting the summation as an infinite sum, we have

$$V^*(\mathcal{D}) = \sup_{\pi \in \Pi} \mathbb{E}^\pi \left[ \sum_{n=1}^{\infty} r(\mathcal{D}_{n-1}, \mathcal{D}_n) \, \mathbf{1}_{\{n \leq N_B\}} \, \middle| \, \mathcal{D}_0 = \mathcal{D} \right]. \tag{2}$$

A truncated version of (2) that is useful within our scenario tree optimization approach described in Section 4.2 is given by

$$V_N(\mathcal{D}) = \sup_{\pi \in \Pi} \mathbb{E}^\pi \left[ \sum_{n=1}^{N} r(\mathcal{D}_{n-1}, \mathcal{D}_n) \, \mathbf{1}_{\{n \leq N_B\}} \, \middle| \, \mathcal{D}_0 = \mathcal{D} \right], \tag{3}$$

where $N$ is a fixed number of "look-ahead steps." When for each $c$ drawn from the prior, there exists a lower bound on $c(x)$ over $\mathbb{X}$ so that (1) is well-defined, it follows that the truncation becomes more accurate when $N$ becomes large: $\lim_{N \to \infty} V_N(\mathcal{D}) = V^*(\mathcal{D})$. This serves as the motivation for our acquisition function described below.

## 4   Budgeted Multi-Step Expected Improvement

We now derive the *budgeted multi-step expected improvement* (B-MS-EI) acquisition function, where the main idea is to solve the truncated problem given in (3). Accordingly, the acquisition function is parameterized by $N$, the maximum number of look-ahead steps. In Section 4.2, we

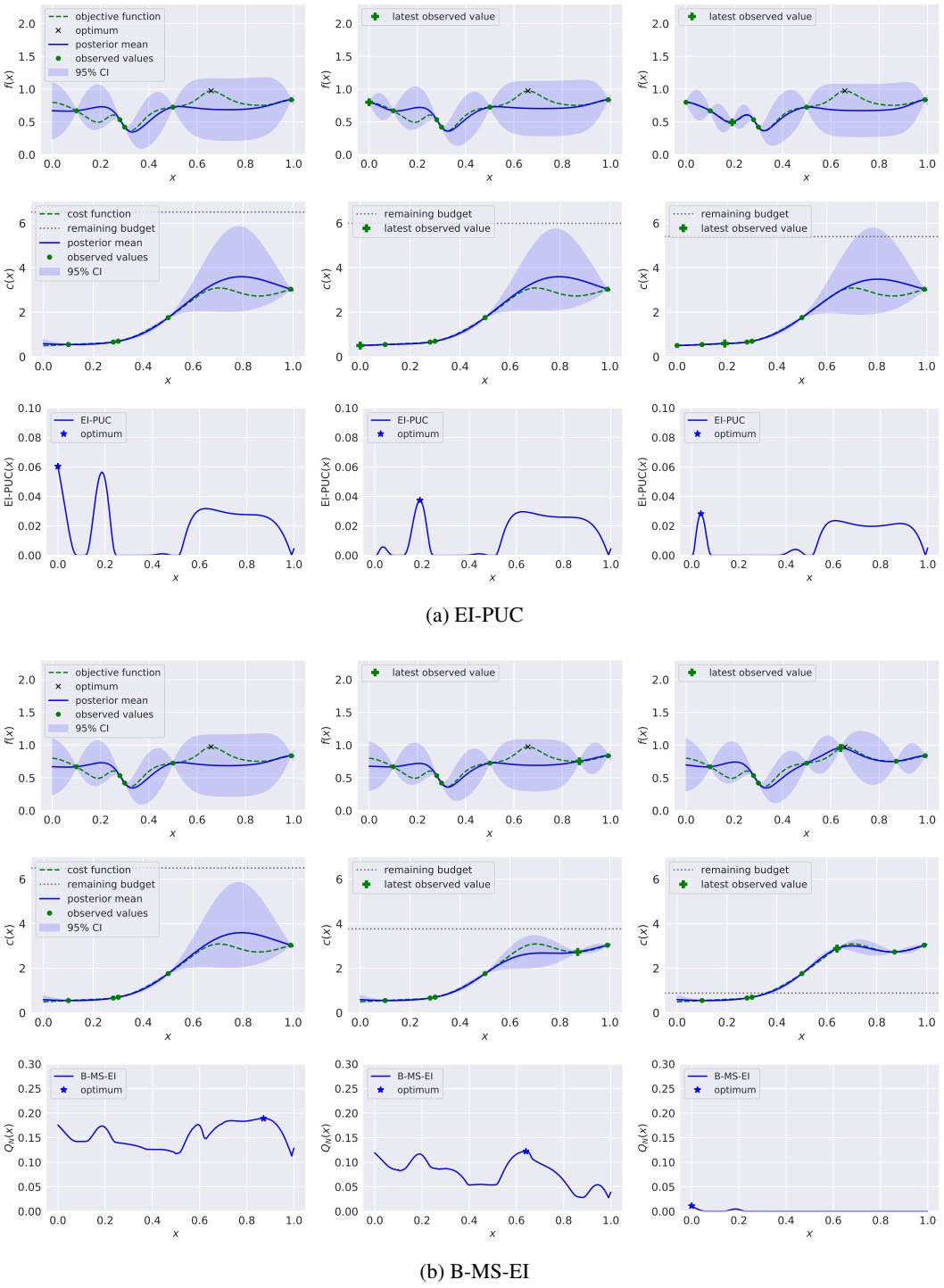

Figure 2: The top six panels show the EI-PUC strategy and the bottom six show our B-MS-EI strategy. In each group, the top and middle rows show the posterior on the objective and cost, respectively, and the bottom row shows the implied acquisition function. Time moves from left to right, where the first column shows the initial posteriors, and the second and third columns show the posteriors after one and two evaluations. EI-PUC evaluates low-variance, low-mean points that are low-cost but unlikely to reveal values near the global optimum. In contrast, B-MS-EI discovers a point near this global optimum within budget after two evaluations. While the remaining budget of EI-PUC after two evaluations is still relatively large, we include additional plots in Section G of the supplementary material showing that indeed B-MS-EI achieves a better objective value within budget.

discuss how to approximately optimize this acquisition function using a scenario tree, leveraging the technique developed in Jiang et al. (2020b), and in Section 4.3, we discuss how to set the budget that defines our acquisition function when the actual remaining budget is too large to be meaningfully taken into account by a computationally feasible number of look-ahead steps.

## 4.1 Dynamic Programming on the Truncated Problem

Our derivation of B-MS-EI starts with applying Bellman recursion to (3). To this end, we define the one-step marginal value of a measurement at point $x$ given an arbitrary set of observations $\mathcal{D}$ (i.e., a state-action value function, or $Q$-function) to be

$$Q_1(x \mid \mathcal{D}) = \mathbb{E}_{y,z} \left[ r(\mathcal{D}, \mathcal{D} \cup \{(x,y,z)\}) \mathbf{1}_{\{s(\mathcal{D})+z \leq B\}} \right] = \mathbb{E}_{y,z} \left[ (y - u(\mathcal{D}))^+ \, \mathbf{1}_{\{s(\mathcal{D})+z \leq B\}} \right].$$

Proposition 1 below shows that, when $f$ and $\ln c$ are drawn from independent GPs, $Q_1$ admits an analytic expression similar to the one of constrained expected improvement (Schonlau et al., 1998; Gardner et al., 2014). The proof of this result can be found in Section B of the supplementary material.

**Proposition 1.** *Suppose that $f$ and $\ln c$ follow independent Gaussian process prior distributions and that $\mathcal{D}$ is an arbitrary set of observations. Define $\mu_{\mathcal{D}}^f(x) = \mathbb{E}[f(x) \mid \mathcal{D}]$, $\mu_{\mathcal{D}}^{\ln c}(x) = \mathbb{E}[\ln c(x) \mid \mathcal{D}]$, $\sigma_{\mathcal{D}}^f(x) = \mathrm{Var}[f(x) \mid \mathcal{D}]^{1/2}$, and $\sigma_{\mathcal{D}}^{\ln c}(x) = \mathrm{Var}[\ln c(x) \mid \mathcal{D}]^{1/2}$. Then,*

$$Q_1(x \mid \mathcal{D}) = \mathrm{EI}^f(x \mid \mathcal{D}) \, \Phi(\zeta) \, \mathbf{1}_{\{s(\mathcal{D}) \leq B\}}$$

*where $\mathrm{EI}^f$ is the classical expected improvement computed with respect to $f$, $\zeta = \big( \ln(B - s(\mathcal{D})) - \mu_{\mathcal{D}}^{\ln c}(x) \big) / \sigma_{\mathcal{D}}^{\ln c}(x)$, and $\Phi$ is the standard normal cdf.*

In contrast with homogeneous-cost non-myopic formulations, the indicator $\mathbf{1}_{\{s(\mathcal{D}) \leq B\}}$ truncates our reward sequence at the random time before the budget is first depleted. Analogously, we can define the $n$-step value function evaluated at $x$, $Q_n(x \mid \mathcal{D})$, as the expected difference in utility after using the optimal policy to evaluate $n$ additional points, among which $x$ is the first of them. Using the Bellman recursion, one can write $Q_n(x \mid \mathcal{D})$ as

$$Q_n(x \mid \mathcal{D}) = Q_1(x \mid \mathcal{D}) + \mathbb{E}_{y,z} \left[ \max_{x \in \mathbb{X}} Q_{n-1} (x \mid \mathcal{D} \cup \{(x,y,z)\}) \right],$$

which holds for all $n$. Our acquisition function B-MS-EI is defined as $Q_N$, meaning that at every step our sampling policy evaluates $x_{n+1} \in \mathrm{argmax}_{x \in \mathbb{X}} Q_N(x \mid \mathcal{D}_n)$.

## 4.2 Optimizing B-MS-EI via Budgeted One-Shot Multi-Step Trees

Maximizing our acquisition function is challenging as, in principle, this requires nested stochastic optimization over a continuous domain. We build upon the multi-step scenario tree approach of Jiang et al. (2020b) and devise an optimization method based on a sample average approximation of $\max_{x \in \mathbb{X}} Q_N(x \mid \mathcal{D}_n)$. In our approach, each scenario is associated with its own decision variable, allowing the problem to be cast as a single deterministic optimization problem over a higher-dimensional domain instead of a sequence of nested stochastic optimization problems. We tackle the higher-dimensional problem using modern tools of automatic differentiation and batched linear algebra operations (Balandat et al., 2020). We begin by noting that, if we apply Bellman's recursion repeatedly, $Q_N$ can be rewritten as

$$Q_N(x \mid \mathcal{D}) = Q_1(x \mid \mathcal{D}) + \mathbb{E}_{y,z} \Big[ \max_{x_2} \big\{ Q_1(x_2 \mid \mathcal{D}_1) + \mathbb{E}_{y,z} [ \max_{x_3} \{ Q_1(x_3 \mid \mathcal{D}_2) + \cdots \} ] \big\} \Big].$$

Now we consider the Monte Carlo approximation of $Q_N(x \mid \mathcal{D})$ given by

$$\widehat{Q}_N(x \mid \mathcal{D})$$
$$= Q_1(x \mid \mathcal{D}) + \frac{1}{m_1} \sum_{j_1}^{m_1} \left[ \max_{x_2^{j_1}} \left\{ Q_1(x_2^{j_1} \mid \mathcal{D}_1^{j_1}) + \frac{1}{m_2} \sum_{j_2}^{m_2} \left[ \max_{x_3^{j_1 j_2}} \left\{ Q_1(x_3^{j_1 j_2} \mid \mathcal{D}_2^{j_1 j_2}) + \cdots \right\} \right] \right\} \right],$$

where $m_i$, $i = 1, \ldots, N-1$, is the number of samples used in step $i$, and the sets of observations are defined recursively by the equations $\mathcal{D}_i^{j_1} = \mathcal{D} \cup \big\{ (x, y_i^{j_1}, z_i^{j_1}) \big\}$ and

$$\mathcal{D}_i^{j_1 \ldots j_i} = \mathcal{D}_i^{j_1 \ldots j_{i-1}} \cup \big\{ (x_i^{j_1 \ldots j_{i-1}}, y_i^{j_1 \ldots j_i}, z_i^{j_1 \ldots j_i}) \big\},$$

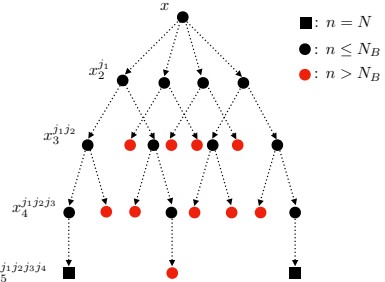

Figure 3: An illustration of the scenario tree representation of our acquisition function for $N = 5$ stages, where $(m_1, m_2, m_3, m_4) = (4, 2, 2, 1)$. Some paths down the tree stop accumulating value earlier than others due to the budget being exhausted, i.e., when $s(\mathcal{D}_i^{j_1 \cdots j_i}) > B$ (ending in red circle nodes). Other paths may terminate due to the maximum look-ahead $N$ before the budget is exhausted (ending in black square nodes), representing some approximation error due to the truncation.

with *fantasy samples* (i.e., samples drawn from the model) $(y_i^{j_1 \cdots j_i}, z_i^{j_1 \cdots j_i}) \sim p(\cdot \mid x_i^{j_1 \cdots j_{i-1}}, \mathcal{D}_i^{j_1 \cdots j_{i-1}})$ obtained via the *reparametrization trick*[2] (Kingma and Welling, 2013; Wilson et al., 2018).

### 4.3 Budget Scheduling via Rollout of Base Sampling Policy

Let $B_n = B - \sum_{i=1}^{n} z_i$ be the remaining budget at time $n$. Since the number of look-ahead steps, $N$, that can be performed in practice is relatively small due to computational limits, $B_n$ may be too large to be taken into account by our acquisition function in a meaningful way, especially during the first few evaluations (if the remaining budget is too large relative to the number of look-ahead steps, then the evaluation costs have little effect in our sampling decisions). Therefore, instead of using the actual remaining budget at step $n$, our acquisition function uses a *fantasy* budget set by a heuristic rule.

We propose a heuristic budgeting rule based on the use of a *base sampling policy* that can be computed quickly. We fantasize $N$ sequential evaluations under this base sampling policy. More specifically, for each of the $N$ steps, we draw fantasy objective and cost values at the point recommended by this policy using the (joint) posterior distribution, then we update the posterior distribution by incorporating these fantasy evaluations, and repeat. The sum of the resulting $N$ fantasy costs determines a cumulative cost incurred by the base sampling policy. Our acquisition function then sets the budget to be the minimum between this cumulative cost and the true remaining budget. This budget is used by our method as if it were the actual budget until depletion, and then the heuristic rule is used again to compute a new fantasy budget. The intuition behind this heuristic is that our acquisition function will try to find the best non-myopic decision using the same budget as the base sampling policy, and thus should perform evaluations that are better or at least as good as those of the base sampling policy. In our experiments, we use EI-PUC-CC as the base policy.

## 5 Experiments

We demonstrate the efficacy of B-MS-EI on four synthetic and four real-world experiments. We report the performance of two variants of our acquisition function, the main variant that uses multiple fantasy samples per step, and the *multi-step path* variant (Jiang et al., 2020b), which is based on a degenerate tree with one fantasy sample per step. For each of these variants, we consider both $N = 2$ and $N = 4$ look-ahead steps. We denote the multi-step path variant with $N$ steps as $N$-B-MS-EI$_p$ and the former simply as $N$-B-MS-EI.

In addition, we report the performance of three baseline acquisition functions from the literature: expected improvement (EI), expected improvement per unit of cost (EI-PUC) (Snoek et al., 2012; Lee et al., 2020b), and expected improvement per unit of cost with cost cooling (EI-PUC-CC) (Lee et al., 2020b). EI-PUC and EI-PUC-CC are currently the de-facto standard approach to cost-aware BO. Since Lee et al. (2020b) assumes that the cost is known (while our experiments are run in the setting of unknown costs), we also integrate with respect to the uncertainty on the cost when computing

---

[2]Note that since we employ Monte Carlo sampling to approximate $Q_N(x \mid \mathcal{D}_n)$, the independence assumption between $f$ and $c$ in Proposition 1 is not necessary for our approach, and one can jointly model $f$ and $c$ (e.g. with a multi-task GP) if there is reason to believe that these functions are correlated, as is often the case in practice. For instance, randomness in training jobs based on variations in test-train splits or weight initialization may interact with adaptive learning rate scheduling and thus affect both model performance and training time.

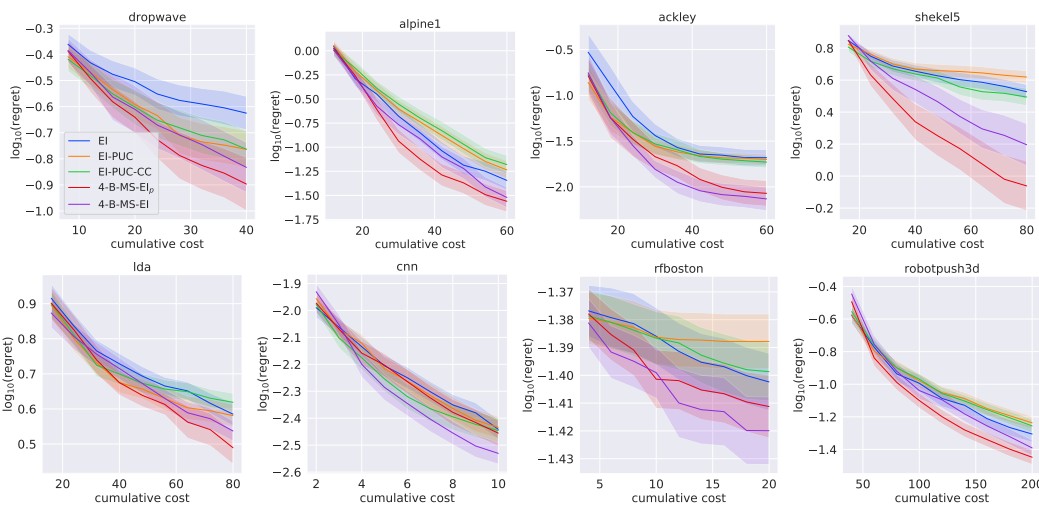

Figure 4: Log-regret of our non-myopic budget-aware BO method (4-B-MS-EI) compared with baseline acquisition functions on a range of problems.

these acquisition functions. Closed form analytical expressions for these acquisition functions can be found in Section C of the supplementary material.

## 5.1 Description of Benchmark Problems

**Synthetic Test Problems.** The first four problems use synthetic objective functions commonly found in the literature: Dropwave, Alpine1, Ackley, and Shekel5 (defined in Section D of the supplementary material). Each objective function is accompanied by a cost function of the form $c(x) = \exp\left[\frac{\alpha}{d} \sum_{i=1}^{d} \cos\left(\beta(x_i - x_i^* + \gamma)\right)\right]$, where $x^*$ is the objective function's maximizer. In each replication we use a different cost function, obtained by sampling $\alpha$, $\beta$, and $\gamma$ uniformly at random over appropriate intervals. Our results thus average over a variety of cost functions. As these parameters vary, cost functions with different characteristics arise. More concretely, $\alpha$ controls the magnitude of the difference between the minimum and maximum costs; $\beta$ controls the variability of the cost; and $\gamma$ controls the cost at the optimum. In particular, if $\gamma = 0$, then $x^*$ is the most expensive point, whereas if $\gamma = \pi$, then $x^*$ is the cheapest point. This family of cost functions emulates a wide range of possible scenarios.

**AutoML Benchmarks.** We consider three AutoML benchmark problems: LDA, CNN, and RF-Boston. The LDA and CNN problems use publicly available data sets from the HPOLib (The HPOlib authors, 2014) and HPOLib1.5 (The HPOlib1.5 authors, 2017) hyperparameter optimization libraries. Following Eggensperger et al. (2018), we use surrogates of the underlying objective and cost functions to emulate the computationally expensive process of training the corresponding models. Details on the construction of the surrogates can be found in Section E of the supplementary material.

The first data set, originally introduced by Hoffman et al. (2010), was obtained by evaluating an online latent Dirichlet allocation algorithm for topic modeling with 3 hyperparameters: mini-batch size $S \in [1, 16384]$ (on a log2 scale), and learning rate parameters $\kappa \in [0.5, 1.0]$ (controls speed at which information is forgotten) and $\tau_0 \in [1, 1024]$ (on a log2 scale, downweights early iterations). These evaluations are expensive and heterogeneous, ranging from 2 to 10 hours each (see Figure 1).

The second data set was obtained by training a 3-layer convolutional neural network on the CIFAR-10 dataset with 5 hyperparameters: "initial learning rate" $\in [10^{-6}, 1.0]$ (on a log10 scale), "batch size" $\in [32, 512]$, and "number of units in layer $k$" $\in [16, 256]$ (on a log2 scale), for $k = 1, 2, 3$. T

Finally, the RF-Boston problem considers optimization of the 5-fold cross validation error when using a random forest (RF) regressor on the Boston dataset, both of which are from sklearn (Pedregosa et al., 2011). We tune the following hyperparameters of the RF regressor: "n_estimators" $\in [1, 256]$ (rounded to the nearest integer), "max_depth" $\in [1, 64]$ (rounded to nearest integer), and "max_features" $\in [0.1, 1]$ (on a log10 scale). The cost of each is evaluation is proportional to the

training time of the model under the evaluated set of hyperparameters (we scale it by training time of a single initial evaluation).

**Energy-Aware Robot Pushing.** We consider a version of the robot pushing problem introduced by Wang and Jegelka (2017), where a robot pushes an object from its origin, $w_{\text{init}} = (0, 0)$, to a target $w_{\text{target}} \in \mathbb{R}^2$. In the our version, we draw 20 different target locations uniformly at random over $\{(w_1, w_2) \in \mathbb{R}^2 : 1 < |w_1|, |w_2| < 5\}$, which are distributed equally across the replications performed. The parameters to be optimized are the location of the robot, $z \in [-5, 5]^2$, and the duration of the push, $t \in [1, 30]$. The objective to minimize is the distance from the object's location after being pushed, $w_{\text{push}}(z, t)$, and the target location, i.e., $f(z, t) = \|w_{\text{target}} - w_{\text{push}}(z, t)\|_2$. The cost function is given by $c(z, t) = \|w_{\text{push}}(z, t)\| + \epsilon$, where $\epsilon$ is a constant, and represents the energy spent by the robot by pushing the object.

## 5.2 Results and Discussion

Figure 4 plots a 95% confidence interval on mean log-regret versus the budget used thus far. We perform experiments using a single budget in each problem; the goal is to minimize regret at the point when the budget is fully exhausted at the right-hand edge of each plot. To focus attention on these budgets, we plot results over the range from 20% to 100% of this overall budget. The results for $N = 2$ look-ahead steps are deferred Section G to the supplementary material to improve readability and also because they are outperformed by the $N = 4$ counterparts for most problems. All implementations use BoTorch (Balandat et al., 2020). The objective and log-cost functions are modeled using independent GPs. Additional details and runtimes can also be found in Section G of the supplementary material. An implementation of our algorithms and numerical experiments can be found at `https://github.com/RaulAstudillo06/BudgetedBO`.

B-MS-EI (red and purple lines) performs favorably compared to existing methods. In some cases (Dropwave and CNN) EI performs worst, while in other cases (Alpine, Shekel, LDA), one of the value-to-cost ratio methods (EI-PUC or EI-PUC-CC) performs worst. B-MS-EI is more computationally intensive to optimize than the standard approaches. The average walltime per acquisition of 4-B-MS-EI, the most expensive variant of our algorithm, ranges from 42 seconds to 7 minutes across the problems, whereas the average walltimes for EI, EI-PUC, and EI-PUC-CC, range from 1 to 30 seconds. However, optimization times on the order of minutes[3] are acceptable when the acquisition function is applied to real-world problems where the function evaluation often takes much longer (e.g., in Figure 1, LDA evaluations require up to 10 hours). In such settings, extra computational time spent optimizing the acquisition function more than pays for itself with improved query efficiency and the savings in objective function evaluation time that results. Nevertheless, there are opportunities to reduce the computational requirements of B-MS-EI; possibilities include re-using the optimized tree for multiple steps or devising better initial conditions for optimization. We leave these for future work.

## 6 Conclusion

We introduced the problem of budgeted BO under unknown and potentially heterogeneous evaluation costs. This arises in a wide range of practical settings where BO is applied. However, the most common paradigm for cost-aware BO, the value-to-cost ratio, fails to account for uncertainty in the cost function as well as the presence of the budget constraint as part of the exploration-exploitation trade-off in a principled way, as we show in Theorem 1. In this work, we provided a dynamic programming formulation of this problem, along with a non-myopic acquisition function that addresses the shortcomings of the existing methods. Our acquisition function is derived by truncating the aforementioned dynamic program, and is approximately maximized following a one-shot multi-step tree approach. Our experiments show that our acquisition function outperforms existing methods on a number of synthetic and real-world examples that exhibit heterogeneity in evaluation costs. In the future, we hope to extend our approach to the multi-fidelity setting, where evaluation costs play an even larger role.

---

[3]These times are not unusual for advanced BO methods; see, e.g., Section 4 of Wu and Frazier (2019).

## Acknowledgments

Authors RA and PF were partially supported by AFOSR FA9550-19-1-0283 and FA9550-20-1-0351.

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
