# Multi-Step Budgeted Bayesian Optimization with Unknown Evaluation Costs: Supplementary Material

**Raul Astudillo**
Cornell University
ra598@cornell.edu

**Daniel R. Jiang**
Facebook
drjiang@fb.com

**Maximilian Balandat**
Facebook
balandat@fb.com

**Eytan Bakshy**
Facebook
ebakshy@fb.com

**Peter I. Frazier**
Cornell University
pf98@cornell.edu

## A   Proof of Theorem 1

**Theorem 1.** *The approximation ratios provided by the EI and EI-PUC policies are unbounded. That is, for any $\rho > 0$ and each policy $\pi \in \{\mathrm{EI}, \mathrm{EI\text{-}PUC}\}$, there exists a Bayesian optimization problem instance (a prior probability distribution over objective and cost functions, a budget, and a set of initial observations $\mathcal{D}_0$) where*

$$V^*(\mathcal{D}_0) > \rho \, V^\pi(\mathcal{D}_0).$$

*Proof.* We prove the result in two parts, first focusing on EI-PUC, and then on EI.

**Part 1: EI-PUC.** To show the result for EI-PUC, we construct a problem instance with a discrete finite domain and no observation noise. Let $\epsilon$ and $\delta$ be two strictly positive real numbers and let $K = \lceil (1+\delta)/\epsilon \rceil$. Suppose the domain is $\mathbb{X} = \{0, 1, \ldots, K+1\}$ and that the overall budget of the problem is $1 + \delta$. The prior on $f(x)$ is independent across $x \in \mathbb{X}$, all points have zero mean under the prior, and the costs $c(x)$ are all known. We assume that the point $x = 0$ has already been measured and has value $f(0) = 0$. The next $K$ points are "low-variance, low-cost" and the final point is "high-variance, high-cost."

- For low-variance, low-cost points, $x \in \{1, \ldots, K\}$, $f(x)$ is normally distributed with mean 0 and variance $\epsilon^2$ and $c(x) = \epsilon$. Thus, $\mathrm{EI}(x)/c(x)$ for these arms is $\mathbb{E}[(f(x) - 0)^+]/\epsilon = \mathbb{E}[\epsilon Z^+]/\epsilon = \mathbb{E}[Z^+]$, where $Z$ is a standard normal random variable.

- For the high-variance, high-cost point, $x = K+1$, $f(x)$ is normally distributed with mean 0 and variance 1, and $c(x) = 1 + \delta$. Thus, $\mathrm{EI}(x)/c(x)$ for these arms is $\mathbb{E}[(f(x) - 0)^+]/(1 + \delta) = \mathbb{E}[Z^+]/(1 + \delta)$.

One feasible policy for the problem is to "measure the high-variance point once." Under this policy, the budget is exhausted after the first measurement and the overall value is $\mathbb{E}[Z^+]$.

Let us consider the EI-PUC policy. By the calculation above, on the first evaluation, it measures a low-variance arm. Then, after that, the remaining budget is strictly less than $1 + \delta$ and it can only measure low-variance, low-cost points. It measures $N = \lfloor (1+\delta)/\epsilon \rfloor$ such points in total (since there is no observation noise, EI of a measured point is zero and a new point is measured each time). We show, in a calculation below, that the value of this policy goes to 0 as $\epsilon \to 0$ while we hold $\delta$ fixed. That is, we show that

$$\lim_{\epsilon \to 0} V^{\mathrm{EI\text{-}PUC}}(\mathcal{D}_0) = 0. \tag{1}$$

35th Conference on Neural Information Processing Systems (NeurIPS 2021).

There are no policies other than the two just described, so the "measure the high-variance point once" policy becomes optimal for all $\epsilon$ sufficiently close to 0. Thus, $\lim_{\epsilon \to 0} V^*(\mathcal{D}_0) = \mathbb{E}[Z^+] > 0$. Thus, given any finite strictly positive $\rho$, there is a $\epsilon$ small enough whose corresponding problem instance satisfies

$$V^*(\mathcal{D}_0) > \rho \, V^{\text{EI-PUC}}(\mathcal{D}_0).$$

To complete the proof for EI-PUC, we now show (1). Under the EI-PUC policy, the values $\{y_n\}_{n=1}^N$ from the sequence of measured points (after the initial point $x = 0$) are independent and identically distributed normal random variables, with mean 0 and variance $\epsilon^2$. The expected value under this policy is then $\mathbb{E}[M]$, where $M = \max\{0, y_1, \ldots, y_N\}$.

Let $t$ be an arbitrary positive real number. Using Jensen's inequality, we obtain

$$\begin{aligned}
\exp\big(\mathbb{E}[tM]\big) &\leq \mathbb{E}\big[\exp(tM)\big] \\
&= \mathbb{E}\big[\max\{1, \exp(ty_1), \ldots, \exp(ty_N)\}\big] \\
&\leq \mathbb{E}\left[1 + \sum_{n=1}^N \exp(ty_n)\right] \\
&= 1 + N \exp(t^2\epsilon^2/2) \\
&\leq (N+1)\exp(t^2\epsilon^2/2),
\end{aligned}$$

where we note in the last step that $\exp(t^2\epsilon^2/2) \geq 1$.

Thus, $\mathbb{E}[M] \leq \ln(N+1)/t + (t\epsilon^2)/2$, and taking $t = \sqrt{2\ln(N+1)}/\epsilon$ we obtain

$$\mathbb{E}[M] \leq \epsilon\sqrt{2\ln(N+1)} \leq \epsilon\sqrt{2\ln\left(\frac{1+\delta+\epsilon}{\epsilon}\right)}.$$

Using L'Hôpital's rule, we see that the right hand side of the above inequality converges to 0 as $\epsilon \to 0$ and, therefore, so does $\mathbb{E}[M]$.

**Part 2: EI.** We now show the result for EI. We consider almost the same setup as in the example above, with the only difference being that the low-cost points now have variance $(1-\epsilon)^2$.

This time $\text{EI}(x) = (1-\epsilon)\,\mathbb{E}[Z^+]$ for the low-cost points and $\text{EI}(x) = \mathbb{E}[Z^+]$ for the high-cost point. Hence, the EI policy evaluates the high-cost point and exhausts the budget after that single evaluation, thus implying that $V^{\text{EI}}(\mathcal{D}_0) = \mathbb{E}[Z^+]$.

Now consider the policy that measures low-cost points only. The expected value under this policy is $\mathbb{E}[M]$, where where $M = \max\{0, y_1, \ldots, y_N\}$, and $y_1, \ldots, y_N$ are independent identically distributed normal random variables with mean 0 and variance $(1-\epsilon)^2$.

We have

$$\mathbb{E}[M] \geq \mathbb{E}\left[\max_{n=1,\ldots,N} y_n\right] \geq (1-\epsilon)\sqrt{a\ln N} \geq (1-\epsilon)\sqrt{a\ln\left(\frac{1+\delta}{\epsilon} - 1\right)},$$

where $a = (\pi \ln 2)^{-1}$ and the second inequality follows by Theorem 1 in Kamath (2015). The expression on the right-hand side of the above inequality goes to $\infty$ as $\epsilon \to 0$. Since $V^*(\mathcal{D}_0) \geq \mathbb{E}[M]$, it follows that, for any $\rho > 0$, there exists $\epsilon$ small enough whose corresponding problem instance satisfies

$$V^*(\mathcal{D}_0) > \rho \, V^{\text{EI}}(\mathcal{D}_0),$$

which completes the proof. $\qquad\square$

## B   Proof of Proposition 1

**Proposition 1.** *Suppose that $f$ and $\ln c$ follow independent Gaussian process prior distributions and that $\mathcal{D}$ is an arbitrary set of observations. Define $\mu_{\mathcal{D}}^f(x) = \mathbb{E}[f(x) \,|\, \mathcal{D}]$, $\mu_{\mathcal{D}}^{\ln c}(x) = \mathbb{E}[\ln c(x) \,|\, \mathcal{D}]$, $\sigma_{\mathcal{D}}^f(x) = \text{Var}[f(x) \,|\, \mathcal{D}]^{1/2}$, and $\sigma_{\mathcal{D}}^{\ln c}(x) = \text{Var}[\ln c(x) \,|\, \mathcal{D}]^{1/2}$. Then,*

$$Q_1(x \mid \mathcal{D}) = \begin{cases} \text{EI}^f(x \mid \mathcal{D})\,\Phi\left(\frac{\ln((B-s(\mathcal{D}))-\mu_{\mathcal{D}}^{\ln c}(x))}{\sigma_{\mathcal{D}}^{\ln c}(x)}\right), & \text{if } s(\mathcal{D}) < B, \\ 0, & \text{otherwise,} \end{cases}$$

*where*

$$\text{EI}^f(x \mid \mathcal{D}) = \Delta(x)\Phi\left(\frac{\Delta_{\mathcal{D}}(x)}{\sigma_{\mathcal{D}}^f(x)}\right) + \sigma^f(x)\varphi\left(\frac{\Delta_{\mathcal{D}}(x)}{\sigma_{\mathcal{D}}^f(x)}\right)$$

*is the classical expected improvement computed with respect to $f$; $\varphi$ and $\Phi$ are the standard normal pdf and cdf, respectively; and $\Delta_{\mathcal{D}}(x) = \mu^f(x) - u(\mathcal{S})$.*

*Proof.* Recall that

$$Q_1(x \mid \mathcal{D}) = \mathbb{E}\left[(f(x) - u(\mathcal{D}))^+ \mathbf{1}_{\{s(\mathcal{D})+c(x)\leq B\}}\right].$$

But, since $f$ and $\ln c$ are assumed to be independent,

$$\begin{aligned} Q_1(x \mid \mathcal{D}) &= \mathbb{E}\left[(f(x) - u(\mathcal{D}))^+\right]\mathbb{E}\left[\mathbf{1}_{\{s(\mathcal{D})+c(x)\leq B\}}\right] \\ &= \text{EI}^f(x \mid \mathcal{D})\mathbb{P}\left(s(\mathcal{D}) + c(x) \leq B\right). \end{aligned}$$

Now note that, if $s(\mathcal{D}) \geq B$, then $\mathbb{P}\left(s(\mathcal{D}) + c(x) \leq B\right) = 0$. If, on the other hand, $s(\mathcal{D}) < B$, then

$$\begin{aligned} \mathbb{P}\left(s(\mathcal{D}) + c(x) \leq B\right) &= \mathbb{P}\left(\ln c(x) \leq \ln\left(B - s(\mathcal{D})\right)\right) \\ &= \Phi\left(\frac{\ln\left((B - s(\mathcal{D})) - \mu_{\mathcal{D}}^{\ln c}(x)\right)}{\sigma_{\mathcal{D}}^{\ln c}(x)}\right), \end{aligned}$$

which concludes the proof. $\qquad\square$

## C  Closed-Form Expressions of EI-PUC and EI-PUC-CC

Recall that the EI-PUC and EI-PUC-CC acquisition functions are defined by

$$\text{EI-PUC}(x \mid \mathcal{D}) = \mathbb{E}_{\mathcal{D}}\left[\frac{\{f(x) - f_n^*\}^+}{c(x)}\right],$$

and

$$\text{EI-PUC-CC}(x \mid \mathcal{D}) = \mathbb{E}_{\mathcal{D}}\left[\frac{\{f(x) - f_n^*\}^+}{c(x)^\nu}\right],$$

respectively, where $\nu$ is the ratio between the remaining budget and the initial budget.

Here we show that, under the same conditions of Proposition 1, EI-PUC and EI-PUC-CC have closed form expressions. This is summarized in Proposition 2 below. Note that this result gives the closed-form expression of EI-PUC as a special case by taking $\nu = 1$.

**Proposition 2.** *Let $\nu$ be an arbitrary positive real number and suppose that the conditions in Proposition 1 are satisfied. Then,*

$$\mathbb{E}_{\mathcal{D}}\left[\frac{\{f(x) - f_n^*\}^+}{c(x)^\nu}\right] = \text{EI}^f(x \mid \mathcal{D})\exp(-\nu\mu_{\mathcal{D}}^{\ln c}(x) + \nu^2\sigma_{\mathcal{D}}^{\ln c}(x)^2/2),$$

*Proof.* Since $f$ and $\ln c$ are assumed to be independent,

$$\begin{aligned} \mathbb{E}_{\mathcal{D}}\left[\frac{\{f(x) - f_n^*\}^+}{c(x)^\nu}\right] &= \mathbb{E}_{\mathcal{D}}\left[\{f(x) - f_n^*\}^+\right]\mathbb{E}_{\mathcal{D}}\left[1/c(x)^\nu\right] \\ &= \text{EI}^f(x \mid \mathcal{D})\mathbb{E}_{\mathcal{D}}\left[\exp(-\nu\ln c(x))\right]. \end{aligned}$$

The well-known formula for the moment generating function of normal random variable gives

$$\mathbb{E}_{\mathcal{D}}\left[\exp(-\nu\ln c(x))\right] = \exp\left(-\nu\mu_{\mathcal{D}}^{\ln c}(x) + \nu^2\sigma_{\mathcal{D}}^{\ln c}(x)^2/2\right),$$

which completes the proof. $\qquad\square$

## D  Synthetic Test Problems Details

Below we provide additional details on the synthetic test problems used in the numerical experiments.

**Dropwave:**

- $f(x) = \left(1 + 12\sqrt{x_1^2 + x_2^2}\right) / \left(0.5(x_1^2 + x_2^2) + 2\right)$.
- $\mathbb{X} = [-5.12, 5.12]^2$.
- $\alpha \in [0.75, 1.5]$, $\beta \in [2\pi/5.12, 6\pi/5.12]$, $\gamma \in [0, 2\pi]$.

**Alpine1:**

- $f(x) = \sum_{i=1}^{d} |x_i \sin(x_i) + 0.1 x_i|$.
- $\mathbb{X} = [-10, 10]^d$.
- $\alpha \in [0.75, 1.5]$, $\beta \in [2\pi, 6\pi]$, $\gamma \in [0, 2\pi]$.
- In our experiment, we set $d = 3$.

**Ackley:**

- $f(x) = 20 \exp\left(-\frac{1}{5}\sqrt{\frac{1}{d}\sum_{i=1}^{D} x_i^2}\right) + \exp\left(\frac{1}{D}\sum_{i=1}^{D} \cos(2\pi x_i)\right) - 20 - \exp(1)$.
- $\mathbb{X} = [-1, 1]^d$.
- $\alpha \in [0.75, 1.5]$, $\beta \in [2\pi, 6\pi]$, $\gamma \in [0, 2\pi]$.
- In our experiment, we set $d = 3$.

**Shekel5:**

- $f(x) = \sum_{j=1}^{5} \left(\sum_{i=1}^{4} (x_i - C_{ij})^2 + b_j\right)^{-1}$, where $b = (1, 2, 2, 4, 4)/10$, and

$$
C = \begin{pmatrix} 4 & 1 & 8 & 6 & 3 \\ 4 & 1 & 8 & 6 & 7 \\ 4 & 1 & 8 & 6 & 3 \\ 4 & 1 & 8 & 6 & 7 \end{pmatrix}.
$$

- $\mathbb{X} = [0, 10]^4$.
- $\alpha \in [0.75, 1.5]$, $\beta \in [2\pi/4, 3\pi/4]$, $\gamma \in [0, 2\pi]$.

## E   AutoML Surrogate Models

The LDA and CNN problems use surrogate models to emulate the computationally expensive process of training the corresponding ML model. The objective and cost surrogate models for CNN are obtained directly from the HPOLib library, and were originally constructed by fitting independent random forest regression models to the objective and cost evaluations over a uniform grid of points. These evaluations are obtained by training a 3-layer convolutional neural network on the CIFAR-10 dataset. For the LDA problem, we fit independent GPs to the objective and log-cost evaluations over a uniform grid of size 288, and then use the resulting posterior means as the surrogate models. These surrogate models can be found at `https://github.com/RaulAstudillo06/BudgetedBO`.

## F   Acquisition Function Optimization

All acquisition functions are optimized as follows. First, we evaluate the acquisition value at $200d$ points over $\mathbb{X}$, and $10d$ of these points are selected using the initialization heuristic described in Appendix F.1 of Balandat et al. (2020). Then, we run L-BFGS-B (Byrd et al., 1995) starting from each point. The point with highest acquisition value of the $10d$ points resulting after running L-BFGS-B is then chosen as the next point to evaluate.

For the variants of B-MS-EI, the $200d$ initial points are chosen using the warm-start initialization strategy described in Appendix D of Jiang et al. (2020). This strategy uses the solution found when optimizing B-MS-EI during the previous iteration and identifies the branch originating at the root of

Table 1: Average per-acquisition runtimes (in seconds) of the algorithms in each problem.

|  | EI | EI-PUC | EI-PUC-CC | 2-B-MS-EI$_p$ | 2-B-MS-EI | 4-B-MS-EI$_p$ | 4-B-MS-EI |
|---|---|---|---|---|---|---|---|
| Dropwave | 4.1 | 5.5 | 6.2 | 24.1 | 26.7 | 42.6 | 87.2 |
| Alpine1 | 7.2 | 9.9 | 10.8 | 36.44 | 44.1 | 106.1 | 236.7 |
| Ackley | 6.4 | 7.3 | 5.7 | 16.6 | 26.0 | 95.8 | 103.5 |
| Shekel5 | 4.5 | 5.5 | 5.6 | 18.3 | 12.8 | 105.8 | 165.5 |
| LDA | 5.4 | 5.9 | 5.5 | 20.7 | 29.6 | 141.9 | 202.1 |
| CNN | 11.7 | 26.9 | 19.3 | 79.3 | 95.0 | 286.1 | 443.2 |
| Robot | 5.9 | 4.0 | 4.7 | 32.2 | 34.3 | 108.2 | 193.7 |
| Boston | 1.8 | 1.6 | 1.5 | 15.7 | 19.7 | 65.1 | 90.5 |

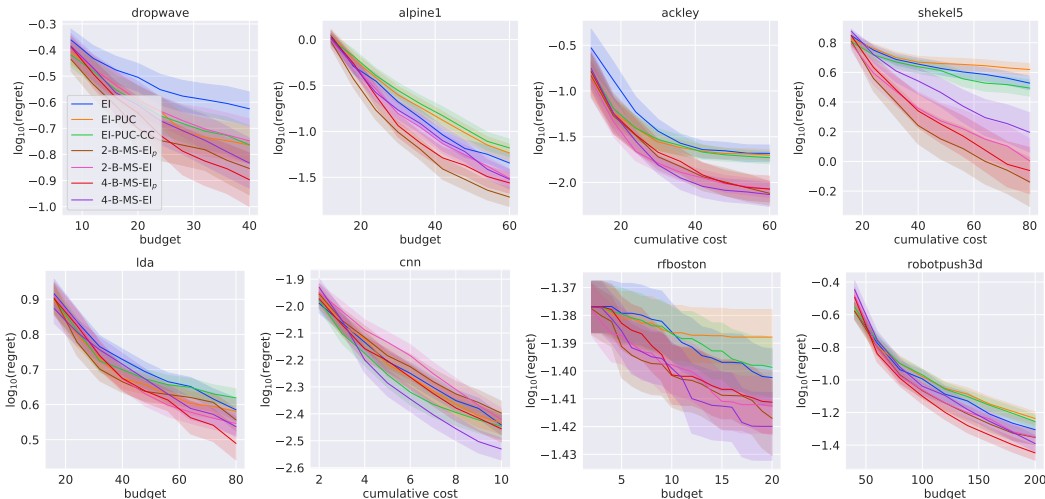

Figure 1: Log-regret of our non-myopic budget-aware BO methods compared with baseline acquisition functions on a range of problems.

the tree whose fantasy sample is closest to the value actually observed when evaluating the suggested candidate on the true function. For the other acquisition functions, the $200d$ initial points are chosen using a Sobol sampling design.

## G   Additional Plots, Initial Design, Hyperparameter Estimation, Runtimes, and Licenses

In each experiment, an initial stage of evaluations is performed using $2(d+1)$ points chosen according to a quasi-random Sobol sampling design over $\mathbb{X}$. Experiments are replicated either 100 (Ackley and Robot Pushing) or 200 (all other test problems) independent trials and we report the average performance and 95% confidence intervals. For all algorithms, the objective function and the log-cost are modeled using independent GPs with constant mean and Matérn 5/2 covariance function. The length scales of these GP models are estimated via maximum a posteriori (MAP) estimation with Gamma priors.

The average runtimes of the algorithms in each problem are summarized in Table 1. Figure 1 plots a 95% confidence interval on mean log-regret versus the budget. In contrast with the plots in the main paper, these plots also include the results for the 2-step variants of our algorithm. Figure 2 below shows the additional evaluations performed by EI-PUC and B-MS-EI within budget in the problem discussed in Figure 2 of the main paper.

The BoTorch package is publicly available under the MIT License. The datasets obtained from the HPOLib and HPOLib 1.5 libraries are publicly available under the GNU General Public License. The source code of the robot pushing problem is publicly available under the MIT License.

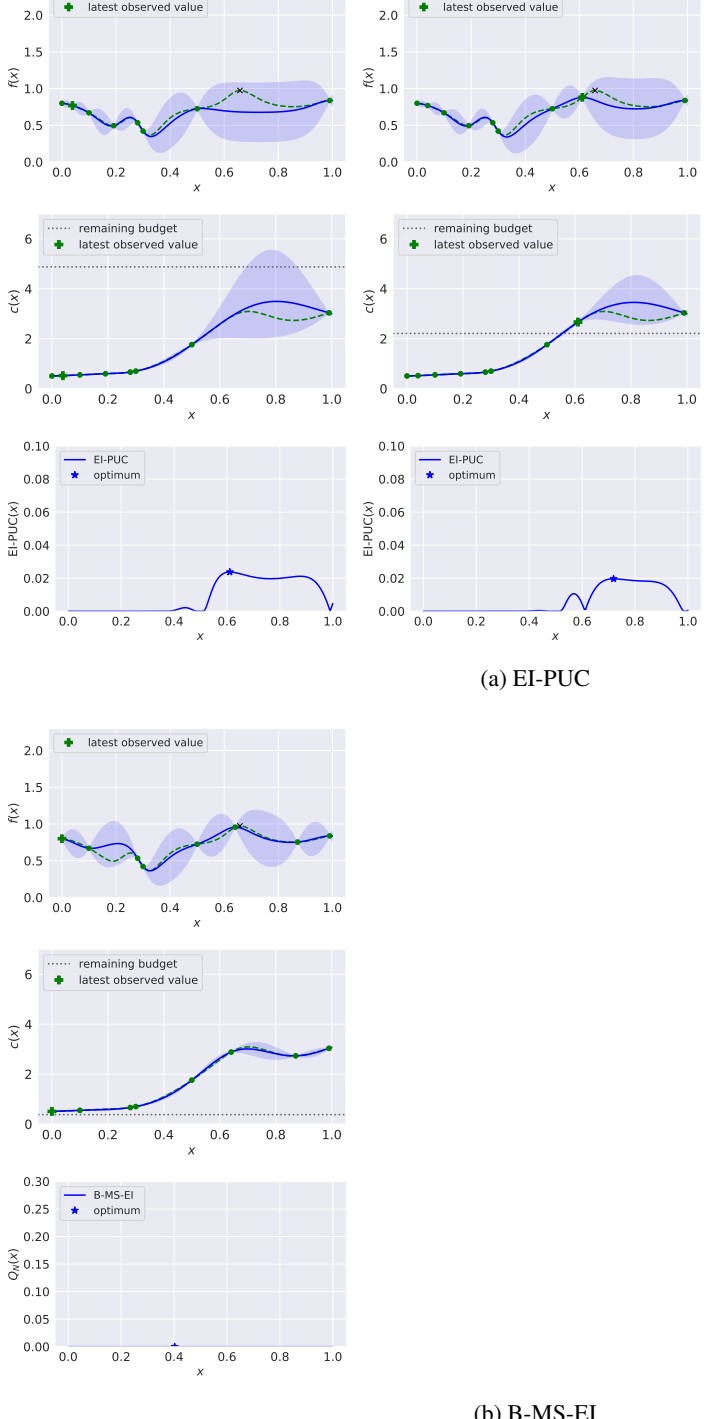

(a) EI-PUC

(b) B-MS-EI

Figure 2: Additional plots showing the evaluations performed within budget by EI-PUC and B-MS-EI. Subsequent evaluations are not plotted because the budget is exhausted after their completion and thus are not taken into account to report performance (i.e., the cost of the next point suggested exceeds the remaining budget). Note that EI-PUC performs two additional evaluations within budget, whereas B-MS-EI performs only one additional evaluation. B-MS-EI achieves a better final performance within budget than the one achieved by EI-PUC.

## H  Budgets Analysis

To understand the effect of the budget on the performance of B-MS-EI, we evaluate its performance in three of our test problems (Dropwave, LDA, and Robot Pushing) using half of the original budget. We also report the performance of EI-PUC-CC, which is the only other benchmark method that is budget-aware. For comparison, we also include the performance of both algorithms under the original budget. The results of this experiment are shown in Figure 3. Remarkably, B-MS-EI seems to benefit from knowing the budget constraint in advance. This does not seem to be the case for EI-PUC-CC, however.

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

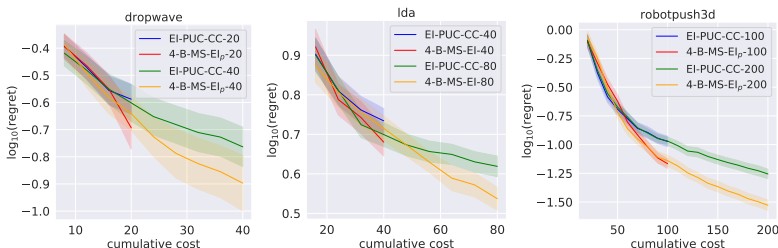

Figure 3: Performance of B-MS-EI (B-MS-EI$_p$ for Dropwave and Robot Pushing) and EI-PUC-CC in three of our test problems under two different budgets. In contrast with EI-PUC-CC, B-MS-EI seems to benefit from knowing the budget constraint in advance.