# OpenReview forum: "Multi-Step Budgeted Bayesian Optimization with Unknown Evaluation Costs"
_NeurIPS.cc/2021/Conference — NeurIPS 2021 Poster_

### Official Review · Reviewer_AFWF · 2021-06-28

**Rating:** 6
**Confidence:** 4

**Summary:**

This paper considers the problem of limited budget Bayesian optimization when the cost of each experiment is unknown and must be estimated by running the experiment. This is an interesting problem introduced in the BO community recently. This paper improves upon previous approaches by treating the problem as a constrained MDP. At each step, the method performs a look-ahead and chooses the next point that maximizes a N-step reward within a budget constraint. The experiments show improved performance over the baselines.

**Limitations And Societal Impact:**

A current limitation is the effectiveness of this method on real large scale optimization problems. The current experiments are on synthetic experiments or relatively small benchmarks. A large scale evaluation on more real world problems is necessary to understand the importance of modeling the evaluation cost in BO. Furthermore, It is understandable that the authors may leave this to future work due to huge computational requirements. However, a discussion about this may be added to the main text.

**Main Review:**

This is an interesting paper providing a method to tackle Bayesian optimization with non-uniform experimental costs. Traditional BO methods treat the experimental cost as a constant for all input designs, however in practice the experimental cost may vary. Previous approaches are myopic in nature, and do not consider the remaining budget. This paper introduces a principled approach to this problem using an MDP formulation and optimize the proposed acquisition function using recently proposed multi-step trees. This approach of incorporating the cost constraints with multi-step trees seems novel. The paper is very well written and the approach is clear.

Pros:
- The problem tackled seems interesting and worthwhile to study. Traditional BO approaches do not consider the true experimental cost, although it may be relevant in certain practical scenarios.
- The approach is principled, using MDPs to model a non-myopic acquisition function, while also constraining the evaluation budget. This is an ideal way of incorporating the cost budget into the decision making process. This is in contrast to previous heuristic approaches that simply divide the original acquisition function by the cost. In particular, the connection to constrained EI looks interesting.
- Experimental results show improved performance over previous state of the art baselines. While there are a few other recently proposed baselines that take the cost into account, they do not work better that the baselines used in the paper.

Cons:
- As described in section 4.3, when the remaining budget is large, the evaluation cost does not affect the optimization. This raises a concern about the practical effectiveness of the method under a large budget (or equivalently, when each experiment incurs a small cost).
- A concise description of the whole approach as an algorithm is missing from the main text.

Some missing references:
- Guinet et. al. Pareto-efficient Acquisition Functions for Cost-Aware Bayesian Optimization
- Abdolshah et. al. Cost-aware Multi-objective Bayesian optimisation
- Paria et. al. Cost-Aware Bayesian Optimization via Information Directed Sampling

Questions:
- Consider a scenario where the budget is high, so a large number of experiments can be performed. Will standard methods (such as EI) outperform in this setting?
- How sensitive is the performance to the parameters of EI-PUC-CC? Were these parameters fixed beforehand for all the experiments?

**Time Spent Reviewing:**

6

---

> ### Author Response · Authors · 2021-08-11
> **Author response to reviewer AFWF**
>
> We would like to thank you for your feedback. We are glad that you found our paper “interesting”, with a problem setup that is “worthwhile to study”, and a “principled” approach that shows “improved performance over previous state of the art baselines”. We respond to your comments and questions below, and look forward to answering any other questions you may have.
>
> $\textbf{Q1:}$ How sensitive is the performance to the parameters of EI-PUC-CC? Were these parameters fixed beforehand for all the experiments?
>
> $\textbf{A1:}$ EI-PUC-CC does not have (hyper)parameters (see equation (3) of Lee et al, 2020). The cost exponent, $\alpha$, is equal to the ratio between the remaining budget and the initial budget. This is stated in line 71 in the paper, and again in Section 5 where, in addition, a closed-form analytical expression of EI-PUC-CC is provided. Thus, the value of $\alpha$ is different for each iteration and depends on the data collected so far; we will make sure to make this more explicit in the revised version of our paper.
>
> $\textbf{Q2:}$ A concise description of the whole approach as an algorithm is missing…
>
> $\textbf{A2:} $Thank you for this suggestion. We will add this in the revised version of our paper.
>
> $\textbf{Q3:}$ ...when the remaining budget is large, the evaluation cost does not affect the optimization… Consider a scenario where the budget is high, so a large number of experiments can be performed. Will standard methods (such as EI) outperform in this setting?
>
> $\textbf{A3:}$ We believe that when the budget is very large relative to the evaluation costs, part of the motivation to use a cost-aware BO method is lost, and thus one might as well use a standard BO method such as EI. However, this situation is rarely true in practice, where the number of evaluations is often severely limited. At the same time, we agree that it would be interesting to understand the performance of our algorithm under larger budgets. To address this concern, we ran two of our numerical experiments, Dropwave and LDA, under twice the evaluation budget used in the paper. For Dropwave, the mean log-regret achieved by 4-B-MS-EI and EI are -1.21 and -0.82, respectively; thus, our method still substantially outperforms EI but the performance gap is slightly smaller than under the original budget.  For LDA, the mean log-regret achieved by 4-B-MS-EI and EI are -0.11 and -0.14, which means that their performance under this larger budget is virtually identical. These results suggest that the performance of our method under moderately large budgets is fairly competitive. Unsurprisingly,  however,  the performance gap between our method and standard BO methods seems to become smaller under larger budgets. We will report a more thorough evaluation of the performance of our method under larger budgets in the revised version of our paper.
>
> $\textbf{Q4:}$ A large scale evaluation on more real world problems is necessary... It is understandable that the authors may leave this to future work due to huge computational requirements. However, a discussion about this may be added to the main text.
>
> $\textbf{A4:}$ The test problems were chosen to exhibit the performance of our algorithm under various realistic conditions that one may find in practice. However, we agree that a larger scale evaluation on more real-world problems would be beneficial and also quite computationally expensive, given that truly real-world problems often require long evaluation times that hinder performing many replications. We will add a discussion of this in the final version of our paper.
>
> $\textbf{Q5:}$ Some missing references…
>
> $\textbf{A5:}$ Thank you for pointing out these papers. We will make sure to include them in the revised version of our paper.

---

> > ### Comment · Reviewer_AFWF · 2021-08-30
> > **Post-rebuttal**
> >
> > Thanks for addressing the concerns. I hope the paper is updated taking into account all the reviewer feedback and concerns.

---

> > > ### Author Response · Authors · 2021-09-01
> > > **Thank you for your confirmation**
> > >
> > > Dear reviewer AFWF,
> > >
> > > We would like to thank you for your confirmation and also again for your valuable feedback. We will make sure to thoroughly take into account all the suggestions and concerns raised by the reviewing team in the revised version of our paper.
> > >
> > > Sincerely,
> > >
> > > The authors

---

### Official Review · Reviewer_tCi1 · 2021-07-16

**Rating:** 7
**Confidence:** 4

**Summary:**

This paper proposes an MDP-style problem formulation for budgeted cost-aware Bayesian optimization and corresponding solutions to solve it. The authors also presented empirical results showing that the proposed method works well in practice.

**Limitations And Societal Impact:**

A key limitation I didn't see mentioned is that users need to select the GP prior for objective and cost functions. Cost function's GP is also much less intuitive given it's in log space.

**Main Review:**

This is a well-written paper. The problem formulation is well thought through, quite reasonable and straightforward to understand. The proposed methods based on dynamic programing and Monte Carlo tree search only seem obvious when the problem formulation is given. The authors did a very good job introducing the problem.

That being said, there are a couple of places that can be further improved to make this paper stronger and help readability.

1, In Theorem 1, only EI related approaches are considered. Is it because EI's acquisition function value is bounded? How does this theorem work for other policies like PI, PI-PUC, UCB, UCB-PUC?

2, In Figure 2, I think it would be helpful to show the acquisition function value in (a) or the corresponding Q function in (b). Right now, it is unclear how either method makes the decision on where to evaluate. Please add this figure in the rebuttal.

3, It would be nice to point out what the reward, state and action mean for the MDP. I think there are some notations that can be improved once this question is sorted out. It seems to me D_n would be the state and x_n the action. But it's unclear to me what reward is. It doesn't seem necessary to introduce an additional D since Eq (1) can just be written as V*(D_0) = sup_\pi E_\pi [...] which also avoids the ambiguity on the variable that the expectation is taken over. Similar for the V functions following that.

4, It would be helpful to write down what V*(D_n) is for any n \in \{1, ..., N\} in addition to just defining V*(D_0). In particular, is there an iterative update rule for V*(D_n) such that it can be computed from V*(D_{n+1})? What would the Q function be in the MDP the authors defined?

5, All the equations in Sec 3.3 look confusing to me not only because of what I mentioned above (unclear definition of V*(D_N) since only V*(D_0) was defined), but also the unnecessary notations like r_N or V_N. I don't think those equations add any clarity. Basically, if I'm understanding correctly, the authors are trying to say they do finite step look ahead but just didn't say it clearly enough. I suggest the authors reword section 3.3 and make the point clear, remove unnecessary equations and follow a concise set of equations.

6, In section 4, it seems to me all the "value functions" actually mean Q functions.




**Time Spent Reviewing:**

3.5

---

> ### Author Response · Authors · 2021-08-11
> **Author response to reviewer tCi1**
>
> We would like to thank you for your detailed feedback. We are glad that you found our paper “well-written”, and sincerely appreciate your comment about doing a “really good job” introducing our problem. We respond to your comments and questions below, and look forward to answering any other questions you may have.
>
> $\textbf{Q1:}$ In Figure 2, I think it would be helpful to show the acquisition function value... Please add this figure in the rebuttal.
>
> $\textbf{A1:}$ Thank you for your suggestion; we agree it would be beneficial to include such figures. We will add them in the revised version of our paper. (We were not able to include them in this response because OpenReview does not support pictures).
>
> $\textbf{Q2:}$ It would be nice to point out what the reward, state and action mean for the MDP…
>
> $\textbf{A2:}$ The state of the MDP is simply the dataset $D_n$, which contains all past objective and cost observations (from $D_n$, one can compute the remaining budget, the best observed value, and the posterior distribution on $f$ and $c$). We will clarify in Section 3.1 that $D_n$ is in fact the state of the MDP. The reward at the $n$-th measurement is the increase in the best observed value after collecting the $n$-th additional point (see line 176). More precisely, if measurements were made at $t=0,\ldots, n$, then the reward after the $n$-th measurement is given by $\max (y_0, \ldots, y_n) - \max (y_0, \ldots, y_{n-1})$. We will clarify this in the revised version of our paper. The action of the MDP is the point to measure in each period:  after observing the current state $D_n$, the decision-maker chooses the next point, $x_{n+1}$, to measure.
>
> $\textbf{Q3:}$ It doesn't seem necessary to introduce an additional D since Eq (1) can just be written as V*(D_0) = sup_\pi E_\pi [...] which also avoids the ambiguity on the variable that the expectation is taken over.
>
> $\textbf{A3:}$ Thank you for pointing this out. We agree. Equation (1) could have been written more clearly by putting $D_0$ on the left-hand side and removing the $D$ in the expectation. We will follow this suggestion and unify the notation throughout the paper.
>
> $\textbf{Q4:}$ It would be helpful to write down what V*(D_n) is for any n \in {1, ..., N} in addition to just defining V*(D_0). In particular, is there an iterative update rule for V*(D_n) such that it can be computed from V*(D_{n+1})? What would the Q function be in the MDP the authors defined?
>
> $\textbf{A4:}$ Yes, we will do so; in fact, $V^*(D_n)$ is defined in the same way as $V^*(D_0)$ because the MDP has an indefinite horizon. To your second question, yes, an iterative update rule exists: Let us adopt the convention that $V^*(D) = 0$ for any datasets D where there is no remaining budget. Then, we have
>
> $V^*(D) =  \max_x E[ r(D, D') * \mathbb{I}(s(D') \leq B) + V^*(D’) ]$,
>
> where $D' = D \cup (x, f(x), c(x))$.
>
> The corresponding Q function looks like this:
>
> $Q^*(D, x) = E[ r(D, D’) *  \mathbb{I}(s(D') \leq B)+ \max_{x'} Q^*(D', x') ]$,
>
> where $D' = D \cup (x, f(x), c(x))$ and $Q^*(D, x) = 0$ whenever the dataset $D$ has no remaining budget.
>
> $\textbf{Q5:}$ ...unnecessary notations like r_N or V_N. Basically, if I'm understanding correctly, the authors are trying to say they do finite step look ahead but just didn't say it clearly enough.
>
> $\textbf{A5:}$ Thank you for pointing this out. Your interpretation is correct: we are formulating the finite-step lookahead here. We will remove the $r_N$ notation and replace it with the definition that “$V^*(D) = 0$ for any datasets $D$ where there is no remaining budget”, as described above. $V_N$, however, is necessary because it defines the value of the finite step problem.
>
> $\textbf{Q6:}$ A key limitation I didn't see mentioned is that users need to select the GP prior for objective and cost functions. Cost function's GP is also much less intuitive given it's in log space.
>
> $\textbf{A6:}$ This is a good point. BO methods in general rely on well-specified prior distributions.  While the cost function is in log space, log transformations have been used with success in the past in BO for modeling functions that are known to be positive (see, e.g., Swersky et al. (2013). Multi-task bayesian optimization). We recommend using cross-validation every few iterations to assess model accuracy. We will add a discussion on this in the revised version of our paper.
>
> $\textbf{Q7:}$ In Theorem 1, only EI related approaches are considered. Is it because EI's acquisition function value is bounded? How does this theorem work for other policies like PI, PI-PUC, UCB, UCB-PUC?
>
> $\textbf{A7:}$ EI-related approaches are usually more sample efficient than ones based on PI and UCB; they are also much more widely used in practice. This is why we focus on EI and EI-PUC in Theorem 1. In addition, EI-PUC has a very clear interpretation of marginal gain in value per effort spent, while PI-PUC and UCB-PUC lack this interpretation. It is possible, however, to prove Theorem 1 for other acquisition functions. For example, consider PI and PI-PUC.  PI measures the arm with the largest value for $P(f(x) > f^* + b)$, for some $b > 0$, while PI-PUC measures the arm with largest $P(f(x) > f^* + b)$ over cost.
>
> The following proof sketch shows that both of these acquisition functions have unbounded approximation ratios.
>
> First, here is an example showing that PI-PUC has an unbounded approximation ratio.
>
> All arms have mean $b$.
>
> Arm 1 (the high-variance, high-cost arm) has variance $1/\epsilon$ and cost $1$.
>
> Arms 2 through $K$ (the low-variance, low-cost arm) have variance $\epsilon^2$ and cost 1/2.
>
> The budget is $1$.
>
> $f^*$ is equal to $0$.
>
> The PI-PUC acquisition function is 1/(2*cost) for all arms, which is $1/2$ for the high-variance arm and $1$ for the low-variance arm. Thus, PI-PUC measures one of the low-variance arms for its first measurement. Then, the remaining budget is not large enough to measure the high-variance arm with a subsequent measurement and so it spends its entire budget on low-variance arms.
>
> We send $\epsilon$ to $0$. In this limit, the expected value of the best arm found by the PI-PUC policy is $b$. In contrast, the expected value of the best arm found by measuring the high variance arm goes to infinity. Thus, the approximation ratio is unbounded.
>
> A similar example shows that PI has an unbounded approximation ratio.
>
> All arms have mean $b$.
>
> Arm 1 (the high-cost arm) has variance $1 + \epsilon$ and cost $1$.
>
> Arms 2 through $K$ (the low-cost arms) have variance $1$ and cost $1/K$.
>
> The budget is $1$.
>
> $f^*$ is equal to $0$.
>
> PI, which measures the arm with the largest $P(f(x) > f^* + b)$, measures the high-cost arm with its first measurement, exhausting the budget.  The optimal strategy, however, measures the $K$ low-cost arms.  Holding $\epsilon$ fixed and sending $K$ to infinity, the expected value of the best solution found by PI is constant while the value of the optimal strategy grows to infinity. Thus, PI’s approximation ratio is unbounded.

---

> > ### Comment · Reviewer_tCi1 · 2021-08-27
> > **Response received**
> >
> > I confirm that I read authors' response and other reviewers' reviews. Almost all reviewers raised clarification issues and I believe those will be further clarified in the paper given what the authors promised. While there are limitations (e.g. scalability, prior selection etc), I still believe this paper is making a concrete step further and I wouldn't blame this paper for not being "perfect". The authors also promised that those limitations will be discussed. Hence I think this paper is a good fit for our conference and I recommend acceptance.

---

> > > ### Author Response · Authors · 2021-08-30
> > > **Thank you for your confirmation and support for our paper**
> > >
> > > Dear reviewer tCi1,
> > >
> > > We sincerely appreciate your confirmation and support for our paper. We are grateful for the helpful feedback and will make sure to address all concerns raised by the reviewing team in the revised version of our work.
> > >
> > > Sincerely,
> > >
> > > The authors

---

### Official Review · Reviewer_eGMU · 2021-07-16

**Rating:** 7
**Confidence:** 3

**Summary:**

This paper aims to develop a BO approach that can handle cost heterogeneity of evaluations while having a budget constraint on the total evaluation cost. The proposed acquisition function follows the MDP setting and extends the general Expected Improvement to a budgeted cost-heterogeneous approach.

**Limitations And Societal Impact:**

Yes

**Main Review:**

The proposed algorithm is theoretically sound. The experiments are sufficient and the results are promising.

Detailed comments:
1.	How effective is the heuristic of budget scheduling via rollout when the modelling of cost function (and accordingly those $N$ fantasy costs) is not reasonably accurate? Let us assume that we are at the start of the optimisation (n<10) and $B_n$ is fairly large. So in that case the heuristic mainly depends on the priors that are used in the GP that models the cost function. Please elaborate on this point.

2.	In Line 218, the $\nu_N$ is written as: $\nu_N(.)=\nu_1(.)+E[..\nu_{1}]$, but I kind of missed the concept of dynamic programming here. I assumed the $\nu$ inside the expectation $E_{y,z}[.]$  should be  $\nu_{n-1}$  (given the equation in line 207). Please correct me if I am wrong. Also, it would be useful to have labels ($ \label{...}$) for these 2 equations.

3.	I assume the $\alpha$ in Theorem 1 is not related to Line 70 (the ratio between the remaining budget and the total budget). In that case, I suggest using a different notation in Theorem 1.

4.	I am a bit confused with the intuition of Theorem 1. Specifically, when $0 < \alpha \leq 1$, what is our takeaway from this Theorem?


**Time Spent Reviewing:**

6

---

> ### Author Response · Authors · 2021-08-11
> **Author response to reviewer eGMU**
>
> We sincerely appreciate your detailed feedback. We are glad that you found our method “theoretically sound” and our empirical analysis “sufficient” and “promising”. We respond to your comments and questions below, and look forward to answering any other questions you may have.
>
> $\textbf{Q1:}$ I assume the $\alpha$  in Theorem 1 is not related to Line 70…
>
> $\textbf{A1:}$ These values are not related. Thank you for pointing it out. We will fix this in the revised version of our paper.
>
> $\textbf{Q2:}$ I am a bit confused with the intuition of Theorem 1. Specifically, when $0\leq \alpha \leq 1$ what is our takeaway from this Theorem?
>
> $\textbf{A2:}$ For $\alpha\leq 1$, the conclusion of the theorem is trivial because, by definition, $V^*(\mathcal{D}_0) \geq V^{\pi}(\mathcal{D}_0)$. However, for $\alpha > 1$, and especially for large values of $\alpha$, this result shows that there exist problem instances where the performance of EI-PUC (and EI) is much worse than the performance of the optimal non-myopic policy. This provides insight into why our method, which is a principled approximation of the optimal non-myopic policy, outperforms EI-PUC (and EI) in numerical experiments.
>
> $\textbf{Q3:}$ In Line 218, $v_N$ is written as: $v_N(\cdot) = v_1(\cdot) + E[..v_1]$... the $v$ inside the expectation $E_{y,z}[\cdot]$ should be $v_{n-1}$  (given the equation in line 207). Please correct me if I am wrong. Also, it would be useful to have labels () for these 2 equations.
>
> $\textbf{A3:}$ The equation in the paper is correct and is obtained by applying the recursion in line 207 repeatedly. Thus, $v_1$ shows up inside the expectation instead of $v_{n-1}$ because $v_{n-1}$ is rewritten as $v_{n-1}(x\mid\mathcal{D}) =  v_1(x\mid \mathcal{D}) + E_{y,z}\left[\max_{x\in\mathbb{X}}v_{n-2}\left(x\mid \mathcal{D} \cup \{(x, y, z)\}\right)\right]$ using the recursion in line 207. This is explained in lines 221-223 where we state “We begin by noting that, if we apply Bellman recursion repeatedly, $v_N$ can be rewritten...”. We will make sure to add labels to these equations in the revised version of work.
>
> $\textbf{Q4:}$ How effective is the heuristic of budget scheduling via rollout when the modelling of cost function) is not reasonably accurate? Let us assume that we are at the start of the optimisation (n<10) and B_n  is fairly large. So in that case the heuristic mainly depends on the priors that are used in the GP that models the cost function.
>
> $\textbf{A4:}$ Yes, the budget used to plan for the first few iterations depends on the prior over the cost function. When only a handful of evaluations of the cost have been performed, this seems to be a necessary property, since there is no other information available beyond these few evaluations and the prior.  We also note that, despite not using a budget rule, other cost-aware BO methods also face a similar challenge when the number of evaluations is small and the cost model depends mostly on the prior. For example, EI-PUC requires an estimate of the cost and may produce poor decisions if this estimate is inaccurate. However, we believe our method does a better job handling this challenge because it considers uncertainty in the cost when it plans, allowing it to make more sophisticated sampling decisions than a method that simply assumes the cost is equal to the posterior mean (as EI-PUC does implicitly). For example, our method understands the risk associated with measuring at a point with highly uncertain costs (in the extreme, the evaluation could consume the entire budget) and will avoid doing so unless other characteristics of this point make it extremely desirable.

---

### Official Review · Reviewer_spuC · 2021-07-19

**Rating:** 6
**Confidence:** 4

**Summary:**

This work proposes a global optimization method for black-box functions, where an evaluation budget is restricted and an evaluation cost is also unknown. By applying a lookahead strategy, the authors introduce a new acquisition function, a budgeted multi-step expected improvement (B-MS-EI). It is based on a popular acquisition function, expected improvement, and it solves an optimization problem under the setting of heterogeneous and unknown evaluation costs in a non-myopic manner. Finally, the authors demonstrate that the proposed method outperforms the baseline, EI-PUC in various experimental circumstances.

**Ethical Concerns:**

I do not have any ethical concerns.

**Limitations And Societal Impact:**

I do not have any comments on limitations and societal impact. Please see "Main Review" for the detailed reviews.

**Main Review:**

This paper solves a very practical scenarios in Bayesian optimization. Most optimization problems are affected by the evaluation costs we do not often know the form of cost function. Thus, this problem setup is interesting and has not developed much yet. However, I have some concerns on the details of algorithm and numerical results. Please see the comments described below and provide a response for them in the rebuttal.

Pros

+ Interesting topic.
+ Well-organized paper.
+ Novel proposed algorithm.

Cons

- Some missing details.
- Weak numerical experiments.

Detailed Comments

1. I would like to see the intuitive explanation of the equation shown in Line 207 (Please add a counter for all equations).
2. Is there any specific way to determine the number of look-ahead steps? As described in Section 5, it is a crucial hyperparameter for the proposed acquisition function. It has to describe in detail.
3. I would like to ask quantitative results on the computational cost of B-MS-EI, compared to other baseline. As described in Section 5.4 as well as the main sections, the computational cost seems like a big burden. It needs to specify in the article.
4. The number of iterations for each Bayesian optimization round is too small; for example, the experiments described in Figure 4 attempt to find a global optimum for 10, 40, 60, or 80 iterations.
5. How many times did you repeat each experiment? I cannot find it in the paper.

Minor issues

* In Line 46, nonmyopic -> non-myopic.
* In Line 96, $\sum_{i = 1}^n c(x_n)$ -> $\sum_{i = 1}^n c(x_i)$.
* In Line 256, Fig 4 -> Fig. 4.

**Time Spent Reviewing:**

5

---

> ### Author Response · Authors · 2021-08-11
> **Author response to reviewer spuC**
>
> We would like to thank you for your feedback. Below, we respond to your questions and comments. We believe that we have fully addressed the concerns stated in the review about the paper (some requests for clarification and a concern about the number of iterations in our experiments that we address below). We thus sincerely hope that this response allows you to consider increasing the rating of our paper.
>
> $\textbf{Q1:}$ I would like to see the intuitive explanation of the equation shown in Line 207.
>
> $\textbf{A1:}$ The equation in Line 207 is the standard Bellman optimality equation, written in the “state-action” form, for the MDP we defined in equation (2). The intuition is as follows. The expected value of $n$ remaining measurements when the first measurement is $x$ (the $v_n(x\mid\mathcal{D})$ term on the left-hand side) is equal to the one-period value of measuring point $x$ (the $v_1(x\mid\mathcal{D})$ term on the right-hand side) plus the maximum value one can attain with $n-1$ remaining measurements (the $E_{y,z}\left[\max_{x\in\mathbb{X}}v_{n-1}\left(x\mid \mathcal{D} \cup \{(x, y, z)\}\right)\right]$ term on the right-hand side). Essentially, the Bellman equation breaks down an $n$-period problem into $n$ one-period problems that one can be solved in a recursive manner (compute $v_1$, then $v_2$, and so on, until $v_n$ is reached).
>
> $\textbf{Q2:}$ Please add a counter for all equations.
>
> $\textbf{A2:}$ We will add a counter.
>
> $\textbf{Q3:}$ How many times did you repeat each experiment? I cannot find it in the paper.
>
> $\textbf{A3:}$ The robot pushing problem was replicated 100 times, and all other problems were replicated 200 times. This and other details about our experimental setup can be found in Section 2 of the supplement. They were deferred to the supplement due to space constraints.
>
> $\textbf{Q4:}$ Is there any specific way to determine the number of look-ahead steps?
>
> $\textbf{A4:}$ Based on our experimental investigation, we recommend setting the number of look-ahead steps to 4 (see lines 260-262). In general, there is a tradeoff between (1) the incremental query efficiency generated by accurately optimizing an acquisition function that looks further ahead; (2) limits on how well one can optimize looking many steps ahead because of limits on computation when optimizing the acquisition function. Our empirical evaluation suggests that looking 4 steps ahead provides a good point along this tradeoff across a wide range of problems.
>
> $\textbf{Q5:}$ I would like to ask for quantitative results on the computational cost of B-MS-EI, compared to other baseline.
>
> $\textbf{A5:}$ The average runtimes of all the methods for each problem can be found in Table 1 of the supplement.
>
> $\textbf{Q6:}$ The number of iterations for each Bayesian optimization round is too small…  for 10, 40, 60, or 80 iterations.
>
> $\textbf{A6:}$ We point out that these values actually denote the budget used and not the number of iterations/evaluations. Since costs are heterogeneous and each method chooses distinct points to evaluate, the number of evaluations performed at the end of the budget varies across methods and replications. For example, for the CNN problem, the budget is equal to 10 hours, and the average number of evaluations performed by EI-PUC and 4-B-MS-EI within this budget are 126 and 61, respectively. For the LDA problem, the budget is equal to 80 hours, and the average number of evaluations performed by EI-PUC and 4-B-MS-EI within this budget are 21 and 20, respectively. While for this second problem the average number of evaluations performed is relatively small, we consider this is realistic for a problem with such expensive evaluations (recall that each evaluation takes between 2 and 10 hours approximately for this problem). We will report this data for all the problems and methods in the revised version of our paper.

---

> > ### Comment · Reviewer_spuC · 2021-08-20
> > **After reading the response**
> >
> > Thank you for your response.
> >
> > Most of my concerns have been resolved.
> >
> > Thus, I slightly increased my score.
> >
> > Please revise a manuscript by considering my concerns, if this paper gets accepted.
> >
> > Best,
> >
> > Reviewer.

---

> > > ### Author Response · Authors · 2021-08-21
> > > **Thank you for considering our response**
> > >
> > > Dear reviewer spuC,
> > >
> > > We sincerely appreciate your response and are very glad that these clarifications allowed you to increase your score. We will make sure to take into consideration your concerns and valuable suggestions in the revised version of our work.
> > >
> > > Sincerely,
> > >
> > > The authors

---

### Decision · Program_Chairs · 2021-09-27

**Decision:**

Accept (Poster)

**Comment:**

The paper studied a variant of the Bayesian optimization problem with additional twists of unknown costs and a budget constraint. All reviewers agree that the problem studied in this paper is practically relevant, the solution (based on dynamic programming and Monte Carlo tree search) is intuitive with rigorous theoretical justifications. Empirical results on several practical (both synthetic and realistic) tasks seem promising.

The authors provided effective feedback during the discussion phase, which helped clarify several concerns (e.g., empirical behavior under large-scale problems under a large budget constraint). The authors are encouraged to take into account the reviews, in particular, to further strengthen the empirical analysis and clarity of the presentation if possible, when preparing a revision.